# Retinoic Acid Induced 1 and Smith–Magenis Syndrome: From Genetics to Biology and Possible Therapeutic Strategies

**DOI:** 10.3390/ijms26146667

**Published:** 2025-07-11

**Authors:** Jasmine Covarelli, Elisa Vinciarelli, Alessandra Mirarchi, Paolo Prontera, Cataldo Arcuri

**Affiliations:** 1Department of Medicine and Surgery, University of Perugia, Piazza L. Severi 1, 06132 Perugia, Italy; cova97@gmail.com (J.C.); alessandra.mirarchi@unipg.it (A.M.); 2Medical Genetics and Rare Disease Unit, Maternal-Infantile Department, S. Maria della Misericordia Hospital, 06132 Perugia, Italy; elisa.vinciarelli@edu.unife.it (E.V.); paolo.prontera@ospedale.perugia.it (P.P.); 3Department of Translational Medicine and for Romagna, University of Ferrara, Via L. Borsari 46, 44121 Brescia, Italy

**Keywords:** HD, RAI1, SMS, PTLS

## Abstract

Haploinsufficiency disorders are genetic diseases caused by reduced gene expression, leading to developmental, metabolic, and tumorigenic abnormalities. The dosage-sensitive Retinoic Acid Induced 1 (*RAI1*) gene, located within the 17p11.2 region, is central to the core features of Smith––Magenis syndrome (SMS) and Potocki––Lupski syndrome (PTLS), caused by the reciprocal microdeletions and microduplications of this region, respectively. SMS and PTLS present contrasting phenotypes. SMS is characterized by severe neurobehavioral manifestations, sleep disturbances, and metabolic abnormalities, and PTLS shows milder features. Here, we detail the molecular functions of *RAI1* in its wild-type and haploinsufficiency conditions (RAI1+/−), as studied in animal and cellular models. RAI1 acts as a transcription factor critical for neurodevelopment and synaptic plasticity, a chromatin remodeler within the Histone 3 Lysine 4 (H3K4) writer complex, and a regulator of faulty 5′-capped pre-mRNA degradation. Alterations of RAI1 functions lead to synaptic scaling and transcriptional dysregulation in neural networks. This review highlights key molecular mechanisms of RAI1, elucidating its role in the interplay between genetics and phenotypic features and summarizes innovative therapeutic approaches for SMS. These data provide a foundation for potential therapeutic strategies targeting RAI1, its mRNA products, or downstream pathways.

## 1. Introduction

Haploinsufficiency disorders (HDs) are a group of genetic conditions caused by the failure of a single functional allele to produce sufficient gene product, leading to a range of developmental, metabolic, or neuropsychiatric abnormalities. These disorders often arise from dosage-sensitive genes, where reduced expression disrupts critical cellular processes. Genes encoding transcription factors, chromatin remodelers, and other regulatory proteins are particularly vulnerable to haploinsufficiency, as even subtle reductions in their expression can have cascading effects on development and homeostasis [1]. Smith–Magenis syndrome (SMS; OMIM #182290) and Potocki–Lupski syndrome (PTLS; OMIM #610883) are rare neurodevelopmental disorders that underscore the critical role of dosage-sensitivity in the *RAI1* gene in determining distinct clinical outcomes [2]. These conditions are caused by reciprocal chromosomal rearrangements in the 17p11.2 region: SMS results from a microdeletion, whereas PTLS is due to a microduplication of the same locus. While SMS is characterized by severe intellectual disability, sleep disturbances, self-injurious behaviors, and obesity, PTLS presents with milder phenotypes, including developmental delay, hypotonia, and anxiety. The contrasting clinical features underscore the pivotal role of the dosage-sensitive *RAI1* gene in these syndromes. *RAI1* plays a pivotal role in multiple cellular processes essential for neuronal function [3].

RAI1 contributes to RNA polymerase II (RNAPII)-mediated transcription termination, collaborating with Rat1 and Rtt103 to ensure the proper dissociation of RNAPII from DNA [4,5]. This process is crucial for maintaining the integrity of RNA products and genome stability. Disruptions in transcription termination, such as those observed in SMS, lead to errors in gene expression, which can contribute to the neurological phenotypes of the disorder. In addition to transcription termination, RAI1 is involved in mRNA decapping and degradation, specifically by recognizing and processing improperly capped pre-mRNAs. Working together with Rat1, RAI1 removes defective caps and triggers the degradation of faulty RNA, ensuring that only properly processed mRNAs are translated. This role in RNA quality control is vital for maintaining accurate gene expression and its disruption in SMS can further exacerbate the disorder’s cognitive and neurological manifestations. Furthermore, RAI1 acts as a key regulator of H3K4 methylation, a crucial modification in chromatin that controls gene expression, especially in neuronal development [6]. By participating in the H3K4 methylation complex, RAI1 ensures that critical genes for brain development are properly expressed. Mutations in RAI1 disrupt this regulation, which is believed to underlie many of the neurological deficits seen in SMS. In addition to these molecular functions, RAI1 plays a critical role in synaptic plasticity, particularly in regulating synaptic scaling in neural networks [7]. It prevents excessive scaling in naive networks and promotes upscaling when neurons experience prolonged inactivity. This balance is essential for maintaining synaptic stability, and when RAI1 function is impaired, as in SMS, synaptic instability leads to cognitive and behavioral issues. Together, these diverse functions of RAI1 highlight its essential role in neuronal health and development. Disruptions in any of these processes contribute to the complex phenotypes observed in SMS and related disorders.

Understanding the physiological molecular role of *RAI1* is essential to exploring its involvement in SMS. Findings derived from model systems, including SMS fibroblast cell lines and *RAI1+/−* mice, have provided valuable insights into cellular and molecular alterations linked to *RAI1* haploinsufficiency, offering a robust foundation for understanding its role in SMS pathogenesis and for developing therapeutic strategies.

Gene expression and lipidomic analyses in SMS models revealed downregulation of the CLOCK/BMAL1 pathway [8], altered lipid metabolism, autophagic dysfunction, mitochondrial impairment, and increased oxidative stress leading to cell death [9]. RAI1 is expressed in postmitotic neurons from development through adulthood. Conditional knockouts showed that RAI1 loss in specific neurons causes motor, cognitive, and metabolic deficits [9]. RAI1 directly regulates BDNF expression and its depletion in hypothalamic neurons impairs BDNF-TRKB signaling and appetite control [7,10,11,12,13]. These findings underscore the central role of RAI1 in neurodevelopment and cellular homeostasis in SMS.

Although our understanding of SMS and PTLS has advanced significantly since the original description of SMS 38 years ago, effective treatments remain elusive [14]. Recent therapeutic strategies aim to address RAI1 dysfunction directly or modulate its downstream pathways. These strategies include CRISPRa to enhance residual RAI1 expression and small molecules targeting neurotrophic pathways, such as BDNF-TRKB [9,10]. While promising, these approaches face challenges in achieving effective delivery and long-term efficacy, particularly within the central nervous system. An alternative approach could be SINEUPs, which regulate translation via specific long non-coding RNA molecules [15].

In this review, we explore the molecular underpinnings of SMS and PTLS, focusing on the pathophysiological roles of RAI1, its involvement in critical neuronal pathways, and the emerging landscape of gene- and RNA-based therapies. Despite significant progress, the quest for a definitive cure continues, underscoring the need for further research into innovative therapeutic strategies.

## 2. Haploinsufficiency Disorders

Haploinsufficiency disorders (HDs) are a class of genetic diseases associated with intellectual disability, developmental or metabolic disorders, or tumorigenesis. The pathogenic mechanism is due to the haploinsufficiency of a single or multiple genes, leading to reduced transcript levels to approximately half of the normal level. This means that a single copy of these genes is insufficient to produce the normal or wild-type phenotype, hence the name [16,17,18].

This condition is not always pathological; on the contrary, in most cases, a single normal allele is sufficient to achieve the typical phenotype, a condition known as haplosufficiency. At the genomic scale, it is worth noting that most mutant alleles are recessive. This indicates that their loss-of-function variants are not deleterious when heterozygous. Carriers of cystic fibrosis, phenylketonuria, or albinism are suitable examples of this phenomenon [19]. Moreover, in the case of imprinted genes, it is well known that certain cell types, especially neurons, require expression from only one allele to produce specific proteins at levels sufficient for proper function. In such cases, haplosufficiency is essential to ensure normal cellular activity [20]. Conversely, haploinsufficient genes in humans often encode proteins with structural, regulatory, mechanochemical, or other non-enzymatic functions. These genes are typically dosage-sensitive and include those involved in transcriptional regulation, as well as genes that govern cell development, the cell cycle, and metabolism [21].

Determining the molecular basis of a human haploinsufficiency disorder often extends beyond merely identifying a gene with a pathogenic variant. The origin of haploinsufficiency may stem from two distinct molecular mechanisms. The first is a heterozygous deletion or microdeletion of chromosomal regions that span multiple genes (Figure 1); such alterations can give rise to genomic disorders with autosomal dominant inheritance. The second mechanism involves a loss-of-function point mutation, which can lead to monogenic disorders, also with autosomal dominant inheritance. Microdeletion syndromes, also referred to as contiguous gene syndromes, are by nature multigenic, as the deleted region includes the loci of several genes. However, not all of the deleted genes contribute equally to the resulting complex clinical phenotype. In most cases, one or, at most, two genes play a primary role in defining the characteristic features of the specific syndrome.

Microdeletion syndromes, first identified at the molecular level in the late 1980s (e.g., 15q11–q13 deletions in Prader–Willi and Angelman syndromes), were later conceptualized by James Lupski in 1998 as ‘genomic disorders’ [22], referring to conditions caused by structural genomic rearrangements impacting gene dosage. These rearrangements arise due to the presence of specific genomic regions called duplicons, or low copy repeats (LCRs), also known as segmental duplications. These are repetitive sequences, typically ranging from 200 kb to 400 kb in length, with extremely high sequence homology, reaching up to 99%. They are localized in chromosomal regions prone to microdeletions, such as subtelomeric and pericentromeric areas. Human genome sequencing has revealed that duplicons include pseudogenes as well as functional genes coding proteins crucial for intercellular regulation, including interneuronal communication and metabolic pathways.

Due to the presence of LCRs, during meiosis, chromosomes may fail to align perfectly centromere-to-centromere [23,24] (Figure 1). Instead, one chromosome may “slip” or “slide” over the other by 200 kb or 400 kb, either at terminal or interstitial regions, owing to the presence of duplicons. This phenomenon is known as non-allelic homologous recombination (NAHR), and can result in both microdeletions and microduplications [25,26,27]. NAHR can occur within the same chromatid (intrachromatid), between two chromatids of the same chromosome (interchromatid), or between homologous chromosomes (intrachromosomal). It can result in a microdeletion, whereas microduplication can only occur by following the last two types of rearrangements [28]. Deletions and duplications resulting from NAHR require direct repeats as recombination substrates, whereas inverted repeats can lead to inversions [22].

### 2.1. Smith–Magenis and Potocki–Lupski Syndromes

#### 2.1.1. The 17p11.2 Chromosomal Region

The recognition of the 17p11.2 chromosomal region as a hotspot for genomic disorders, including SMS and PTLS, was a gradual process driven by advancements in genomic technologies [29,30]. The 17p11.2 region is implicated in various rearrangements, highlighting its status as a highly unstable area of the human genome, susceptible to both meiotic and mitotic alteration [31,32,33,34]. These unbalanced rearrangements are due to the presence of LCRs. LCRs are widely distributed throughout the human genome. One region notably enriched in them is the pericentromeric 17p11.2 cytoband located on chromosome 17, also known as the SMS region [35] (Figure 2). There are four LCRs, termed LCR17pA, LCR17pB, LCR17pC, and LCR17pD, as well as three large LCRs called SMS-REPs. Depending on their chromosomal positioning, these LCRs are called SMS-REPP (proximal), SMS-REPM (middle), and SMS-REPD (distal). These repeats are paralogous gene clusters and pseudogenes [36,37] (Figure 2).

The SMS-REPP spans approximately 256 kb and is oriented in the same direction as the SMS-REPD, which is shorter at around 176 kb. The SMS-REPM, about 241 kb, is inverted relative to SMS-REPP and SMS-REPD. This unique architecture may explain why common SMS deletions occur between the proximal and distal SMS-REPs. Although genomic instability in 17p11.2 is also caused by the presence of AT-rich regions and Alu elements, which mediate non-homologous end joining (NHEJ), this structural anomaly occurs consistently [38,39]. The presence of all these sequences makes the region highly prone to imbalance phenomena that can lead to both microdeletions and microduplications. Imbalances in the 17p11.2 cytoband are associated with genetic disorders known as contiguous gene syndromes, as 17p11.2 harbors around 80 genes. Although this suggests a multigenic syndromic condition, not all 80 deleted or duplicated genes equally contribute to the resulting complex phenotype. Generally, one or at most two genes are directly responsible for the specific clinical traits that characterize the syndrome, typically those involved in transcriptional regulation, protein homeostasis, and synaptic brain functions; in other words, dosage-sensitive genes (Figure 2).

#### 2.1.2. Divergent Outcomes in 17p11.2 Imbalances

The *RAI1* gene is located on chromosome 17, spanning approximately 55.5 kb, from position 17,681,458 to 17,811,453 on the GRCh38 assembly. Lipid and metabolic disorders, behavioral problems, sleep disturbances, and cognitive impairments represent contrasting and mirrored clinical features of SMS and PTLS, underscoring the exquisite sensitivity of cells to RAI1 in regulating key biological processes [40].

PTLS patients typically exhibit significantly reduced body weight and height, along with decreased total cholesterol and LDL levels. In stark contrast, most SMS patients develop hypercholesterolemia, elevated LDL levels, hyperphagia, and obesity by adolescence, emphasizing the crucial role of copy number variations in body weight regulation [41]. With regard to the sleep–wake cycle, the majority of SMS patients exhibit an inversion of melatonin rhythm, whereas, PTLS patients tend to experience milder disturbance, primary related to breathing and sleep apnea [20,42,43,44].

It is widely acknowledged that a reduction in gene expression, by approximately 50%, tends to have more severe and far-reaching consequences in complex organisms than overexpression [45,46,47,48]. This principle is exemplified by the contrasting phenotypic outcomes associated with imbalances in the 17p11.2 region. In cases of 17p.11.2 duplication, PTLS patients typically present milder facial dysmorphia and behavioral problems, including atypicality, withdrawal, anxiety, and inattention. In contrast, 17p11.2 deletion in SMS patients results in a more conspicuous and severe clinical profile, characterized by neurological and psychiatric features such as infantile hypotonia, intellectual disability, sleep disturbance, EEG abnormalities and epilepsy, obesity, peripheral neuropathy, and self-injurious and aggressive behaviors [49,50]. Such evidence underscores the disproportionate severity of microdeletions compared to microduplications [51,52]. Microdeletions more significantly disrupt the dosage-sensitive balance of gene expression, triggering cascading effects on developmental and physiological processes. Conversely, microduplications are generally better tolerated, and the associated phenotypes are often mild or subclinical, frequently resulting in underdiagnosis. As a result, many PTLS cases are not identified until the genetic alteration is transmitted to the next generation, where variable expressivity and incomplete penetrance may give rise to a more severe and clinically apparent phenotype. This phenomenon further highlights the clinical subtleties and diagnostic challenges associated with these syndromes.

#### 2.1.3. SMS and PTLS Epidemiology and Clinical Features

SMS is a rare genetic disorder with a prevalence rate of approximately 1 in 2500 individuals worldwide. It affects both sexes equally [53]. Patients diagnosed with SMS exhibit distinctive craniofacial features, including brachycephaly with a broad face, frontal bossing, synophrys, upslanting palpebral fissures, deep-set eyes, a depressed nasal bridge, a short and broad nose, low-set and/or malformed ears, a thin and arched vermilion upper lip, and prognathism; these features often become more pronounced with age. Additionally, brachydactyly with broad and short hands and feet, short stature, and a tendency towards overweight are commonly observed [54]. A characteristic hoarse, low-pitched voice is often noted and may aid in diagnosis [55,56].

Given the significant role of the *RAI1* gene, its haploinsufficiency has multisystemic effects. One of the hallmark features is the disruption of the circadian rhythm, as early studies have shown: melatonin secretion is inverted, with peak levels occurring during the day instead of at night. Dermatological manifestations often include xerosis and folliculitis of the back. Some individuals with SMS also exhibit lighter hair and skin pigmentation compared to their unaffected family members. Dental development is frequently impaired. Cardiac anomalies, including atrial and ventricular septal defects, and tetralogy of Fallot, are also common. Genitourinary malformations, such as unilateral renal agenesis, are frequently reported. Otorhinolaryngological involvement is prominent in SMS patients, with hearing loss in at least one ear occurring in 79% of cases. Hyperacusis and recurrent otitis media (both acute and chronic) are common in children. Orofacial hypotonia contributes to feeding difficulties. Strabismus, myopia, and astigmatism are frequently reported. Impaired antibody production occurs in 60% of individuals, predisposing them to recurrent respiratory infections. Peripheral neuropathy is present in approximately 75% of patients, resulting in an abnormal gait (e.g., foot slapping or toe walking). Reduced sensitivity to temperature and pain can exacerbate the consequences of self-injurious behavior and polyembolokoilomania [57,58]. Epidemiological studies have shown that the 17p11.2 deletion has a greater impact on systemic anomalies such as cardiac, renal, and auditory abnormalities. In contrast, monogenic mutations in *RAI1* have a more significant effect on behavioral and psychological aspects, including hyperphagia, polyembolocoilamania, self-hugging, trichophagia, and self-injurious behaviors [59,60,61,62,63].

Concerning PTLS, patients exhibit a broad spectrum of clinical features, including mild dysmorphic facial traits, hypermetropia, infantile hypotonia, delayed psychoverbal and motor development, intellectual disability of varying severity, autism spectrum disorders, behavioral anomalies, sleep apnea, and cardiovascular defects. However, the overall presentation of PTLS is generally milder and less conspicuous compared to SMS [64,65,66]. This subtler clinical profile, combined with the absence of pathognomonic signs that are obligatory for diagnosis, significantly complicates the recognition of PTLS. The lack of hallmark features often leads to underdiagnosis or misdiagnosis, particularly in cases where the symptoms do not strongly suggest a chromosomal abnormality [67]. As a result, many PTLS cases remain unidentified, highlighting the critical need for heightened clinical awareness and the integration of genomic diagnostic tools to improve detection rates for this underrecognized syndrome.

## 3. Retinoic Acid Induced 1 (RAI1)

### 3.1. Genetics

RAI1 (initially named Gt1) was first discovered as a gene activated in the mouse embryonic tumor cell line P19 by retinoic acid, a treatment that induces the cells to develop into neuronal and glial cells [68]. A few years later, in 1995, RAI1 was recognized as a key regulator of neural and glial cell development in a mouse model of an embryonic tumor cell line following exposure to high levels of retinoic acid [68]. Genetic alterations leading to reduced RAI1 function, first identified by Slager et al., 2003, are linked to a condition that shares similarities with, but is not identical to, the 17p11.2 microdeletion syndrome [38].

*RAI1* (OMIM #607642, NM_030665) is described as having six exons, of which four are coding. The translation start site is located in exon 3 and encodes the majority of the protein (97%) (Figure 2), which is also a mutational hotspot. In the promoter region, it presents many binding sites, including one for retinoic acid, hence the name. Specifically, *RAI1*, has an open reading frame and its main transcript (NM_030665) is the most highly conserved among various splice variants.

In addition to the canonical form, several other isoforms of RAI1 exist, each exhibiting variations, particularly in their untranslated regions (UTRs) or shorter coding sequences. These isoforms are primarily generated through alternative splicing. Isoform 1 corresponds to the full-length canonical sequence, comprising 1906 amino acids, and is predominantly expressed in brain and heart tissues [69]. Three additional isoforms are annotated in the UniProt database, each differing in amino acid composition and domain structure. Isoform 2 is the most similar to the canonical form, containing 1862 amino acids and retaining conserved functional domains. Isoform 3, consisting of 1640 amino acids, lacks the second polyserine tract and the plant homeodomain (PHD), potentially affecting its transcriptional regulatory function [70]. Isoform 4 is the shortest, comprising only 966 amino acids, approximately the first half of the canonical protein. This truncated version lacks nuclear localization signals (NLSs) and resembles aberrant RAI1 forms observed in certain Smith–Magenis syndrome (SMS) patients. While isoforms 1–3 have been primarily identified in brain tissue, isoform 4 has also been detected in muscle and fibroblasts [71,72].

Microdeletions are unbalanced chromosomal structural abnormalities that can occur in terminal or interstitial regions of the chromosome. The size of the deletion is variable and directly proportional to the severity of the phenotype. Partial chromosomal deletions are predominantly observed as de novo events, where a child affected by this condition is born to two apparently healthy parents. The recurrence risk is minimal, with the exception of the rare possibility that one of the parents carries germline mosaicism, which increases the recurrence probability to approximately 1%. SMS is inherited in an autosomal dominant manner. Segregation analyses have shown that the deletion often occurs on the maternally derived chromosome [73]. Less commonly, but still possible, one of the parents may carry a balanced reciprocal translocation, in which case the deleted chromosome is the derivative, carrying both the microdeletion and a translocated portion of the other chromosome originally involved in the translocation [74].

In approximately 90% of cases, SMS is diagnosed based on the presence of a microdeletion within the 17p11.2 cytoband of chromosome 17. The typical size of this deletion is about 3.7 Mb, although smaller deletions (as small as 650 kb) and larger ones (up to 9 Mb) have also been documented. This condition is classified as a contiguous gene syndrome because the 17p11.2 region harbors approximately 80 genes. Among these, the genes associated with Mendelian disorders in humans (OMIM-related) and those that are most significant and extensively studied are highlighted in Figure 3.

The haploinsufficiency of the *RAI1* gene is considered the primary driver of the hallmark symptoms of SMS. However, differences in clinical presentation have been observed between individuals with 17p11.2 deletions and those with isolated point mutations in RAI1. This suggests that the phenotypic variability may stem from the contribution of additional genes within the deleted region [75]. These genes likely play a role in modulating the diverse and severe manifestations observed in deletion cases, underscoring the complexity of the genetic architecture of SMS.

A study examining the relationship between genetic variants and physical characteristics in individuals with deletions of varying sizes suggested that the region between SREBF1 and SHMT1 may contribute to short stature, while hearing loss could be influenced by reduced function of LLGL1, FLCN, and MYO15A (Figure 3). Notably, mutations in MYO15A, which are known to cause non-syndromic deafness (DFNB3; OMIM #600316), have been identified in SMS patients with hearing impairment [3].

Several other genes within the deleted region have been linked to various disorders and the loss of one copy of these genes due to the deletion may unmask the presence of recessive genetic variants, potentially leading to the coexistence of two distinct genetic conditions in the same individual [62]

Mutations in TNFRSF13B (Figure 3) are associated with IgA deficiency-2 (OMIM #609529) and common variable immunodeficiency 2 (OMIM #240500) and may contribute to the IgA deficiency observed in some SMS patients [76,77].

*FLCN*, a tumor suppressor gene located within the 17p11.2 chromosomal region, shares its locus with RAI1. Haploinsufficiency of *FLCN* is typically associated with Birt–Hogg–Dubé syndrome (BHD, OMIM #135150), a monogenic disorder caused by point mutations in *FLCN* leading to lung cysts, spontaneous pneumothorax, and kidney and skin tumors. The typical 17p11.2 deletion in SMS patients can also result in clinical features of BHD [78].

Interestingly, a recent report described a 58-year-old woman with SMS who developed bilateral kidney tumors [79]. Moreover, a recent investigation described four SMS patients exhibiting hallmark features of BHD, including renal cell carcinomas and/or cutaneous fibrofolliculomas, attributed to *FLCN* haploinsufficiency [80]. This overlap is explained by the contiguous gene deletion encompassing not only *RAI1* but also *FLCN*, resulting in haploinsufficiency of both genes. Recent reports have described SMS patients with renal cell carcinomas and cutaneous fibrofolliculomas, hallmark features of BHD, linked to the deletion of *FLCN* in addition to *RAI1*. These findings highlight the broader implications of microdeletions in 17p11.2, where the clinical phenotype of SMS may extend beyond the core symptoms driven by *RAI1* haploinsufficiency to include additional features stemming from the deletion of contiguous genes, like *FLCN*. This underscores the complexity of SMS as a contiguous gene syndrome, where the phenotypic spectrum is shaped by the combined effects of haploinsufficiency of multiple genes within the deleted region, illustrating how the loss of neighboring genes contributes to the variability and breadth of clinical manifestations.

Mutations in the *ALDH3A2* gene cause Sjögren–Larsson syndrome (OMIM #270200), an inherited neurological disorder characterized by dry skin, intellectual disability, spastic muscle stiffness, vision problems, and abnormalities in the white matter of the brain [81]. It has been noticed that individuals with SMS due to deletions are more likely to experience dry skin compared to those with mutations, suggesting a potential role of this gene in this particular symptom [3,82].

For the remaining 10% of patients, where there is strong clinical suspicion but no detected copy number variants, the sequencing of the *RAI1* gene is conducted as a second-tier investigation, as SMS can also present as a monogenic condition [75]. In cases where RAI1 point mutations are identified, 50 likely pathogenic or pathogenic variants are registered in the HGMD. Of these 50 variants, 47 are nonsense mutations, where a codon is replaced by a stop codon, and only 3 are missense variants, resulting in the substitution of a canonical amino acid with one of different size and polarity [75].

### 3.2. Biological Functions

#### 3.2.1. Retinoic Acid Induced 1 Protein Structure

The RAI1 protein is a large protein comprising 1.906 residues and several functional domains (Figure 4).

Starting from the N-terminus, it features a polyglutamine (Poly-Q) tract, a polyserine (Poly-S) tract, and a nuclear localization signal (NLS) that facilitates its nuclear import after cytosolic translation. Following another poly-S tract is a nucleosome-binding domain (NBD), which interacts with nucleosomes both in vitro and in vivo in mouse models [10].

Closer to the C-terminus, a highly conserved plant homeodomain (PHD) motif can be recognized. This PHD motif, characterized by its His-Cys5-His-Cys2-His structure, is essential for chromatin remodeling and binds to the histone H3 tail, earning it the designation of an “epigenetic reader.” Proteomic studies have demonstrated that the RAI1 PHD domain interacts with other transcription factors such as PHF14, TCF20, and HMG20A/iBRAF, suggesting the formation of a “RAI1 complex.” This complex scans chromatin, recognizes unmethylated H3K4, and recruits *KMT2A* (formerly known as *MLL1*), to trimethylate H3K4, thereby promoting downstream gene transcription. These interactions have clinical implications, as RAI1 dysfunction can manifest with symptoms similar to those of Wiedemann–Steiner syndrome, necessitating differential diagnosis.

RAI1 protein levels were found to be altered in the post-mortem prefrontal cortex of patients with schizophrenia, bipolar disorder, and major depression [40]. Moreover, RAI1 expression levels were commonly reduced in several intellectual disability syndromes not directly associated with RAI1 mutations. These include Brachydactyly Mental Retardation Syndrome (BDMR, MIM: 600430), caused by the deletion of the 2q37 chromosomal region containing HDAC4, and 2q23.1 deletion syndrome (MIM: 156200) [8]. This suggests that RAI1 may function as a downstream effector in other neuropsychiatric conditions.

#### 3.2.2. RAI1’s Role in RNAPII-Mediated Transcription Termination in Eukaryotic Cells

A recent and intriguing study by Yanagisawa et al., 2024, establishes a link between RAI1 and the transcription termination complex [4].

In the nucleus, Rat1 and RAI1 combine to form a tetramer that holds one molecule of Rtt103. When recruited by RNAPII, the Rat1-RAI1 tetramer dissociates into dimers. The binding of Rat1-RAI1 to RNAPII requires the prior dissociation of Spt6 and Spt4/5 from RNAPII (with the exception of the KOW5 domain of Spt5). Thus begins the transcription termination process. Rat1 captures the monophosphorylated 5′end of the RNA, generated by the cleavage of the mRNA precursor at the polyadenylation site, thereby initiating the RNA trimming process. Once Rat1 has trimmed the RNA to a length of 20–22 nucleotides, Rat1-RAI1 re-engages RNAPII. Further trimming of the mRNA leads to the collapse of the transcription bubble and the release of the DNA/RNA complex.

Upon completion of transcription termination, Rat1-RAI1 translocate to the now-empty DNA-binding site of RNAPII to occupy it. Alternatively, they may also bind to the RNA exit site of RNAPII to ensure the efficient transfer of the DNA/RNA complex from the RNAPII binding site. This role of Rat1 and RAI1 is crucial for protecting the structure and function of the polymerase until it is recruited into a pre-initiation complex for a new round of transcription.

In this process, RAI1 plays a pivotal role. In collaboration with Rat1 and Rtt103, RAI1 helps ensure the proper dissociation of DNA and RNA from RNAPII, preventing errors in transcription termination and safeguarding genomic stability and the quality of the transcribed RNA [5].

#### 3.2.3. RAI1 in mRNA Decapping and Degradation

Studies have explored the involvement of RAI1 in the recognition of improperly capped pre-mRNAs within the RNA quality control pathway. It is well known that eukaryotic transcripts are initially produced as precursor mRNAs (pre-mRNAs) by RNAP II in the nucleus. These pre-mRNAs undergo multiple processing steps, splicing, 3′-polyadenylation, and 5′-capping, to become fully mature. Once these modifications are complete, the mature mRNAs are exported to the cytoplasm. RAI1 plays a role in mRNA capping quality control by targeting primary transcripts that fail to be capped for degradation [5].

The 5′-cap structure is typically added to nascent pre-mRNAs once approximately 20 nucleotides have been synthesized. This process involves the attachment of a 7-methylguanosine to the first nucleotide of the transcript via its 5′-hydroxyl group, forming a unique 5′-to-5′ triphosphate linkage [83].

Previous studies have demonstrated that RAI1 has enzymatic activity, including pyrophosphatase, decapping, and 5′-3′ exoribonuclease activities, and when it interacts with the Rat1/Xrn2 complex, it associates with the PolII mRNA transcription machinery. Furthermore, studies suggest that RAI1 and Rat1 exhibit allosteric characteristics, as neither enzyme functions efficiently without its respective partner. Specifically, RAI1′s activity, which converts uncapped pre-mRNA 5′ ends into monophosphorylated 5′ ends, is stimulated by Rat1, and vice versa, with Rat1′s exoribonuclease activity also being stimulated by RAI1 [84].

In eukaryotic cells, when a pre-mRNA with an incorrect cap is generated, the cap-binding complex cannot bind, disrupting the interaction with the guard protein Npl3. The substrate is then available to be recognized, although the precise recognition mechanism remains unclear, and is bound by the Rat1–RAI1 complex, which removes the cap and degrades the faulty pre-mRNA in the nucleus in a 5′-3′ direction. It is crucial that this recognition mediated by Rat1–RAI1 occurs efficiently, because effective gene expression requires properly matured mRNAs for accurate translation.

Given the importance of this mechanism, RAI1′s role in mRNA quality control becomes even more critical in the context of diseases like SMS, where disruptions in RNA processing can have profound impacts on gene expression and cellular functions.

#### 3.2.4. RAI1 Key Regulator of H3K4 Methylation Ensures Normal Brain Development

Another activity associated with RAI1, as reported in several studies, is its ability to modulate chromatin [7]. Unlike traditional chromatin regulatory machines, which involve histone-modifying enzymes, RAI1 emerges as a crucial player with a still underexplored role in chromatin regulation. Chromatin modification falls within the field of epigenetics and involves histones, among other factors. Two classes of enzymes are known to be involved in post-translational modifications of histones: “writers” and “erasers,” which generally work together in a coordinated manner [85]. RAI1 is part of the H3K4me writer complex, counterbalancing the LSD1-containing H3K4me eraser complex to ensure normal brain development. The RAI1-TCF20-PHF14 complex functions within the H3K4 writer complex, and MLL1 may also participate in the RAI1 complex through its association with iBRAF. RAI1, along with TCF20 and PHF14, can bind to histone tails and nucleosomes via the PHD domain. However, it remains unknown whether these domains specifically recognize particular histone modifications.

It is also noted that both RAI1 and TCF20 possess another NBD upstream of the ePHD domain (Figure 4). The NBD–nucleosome interaction appears to be independent of the N-terminal tail of histone H3, which contains H3K4, suggesting that the NBD is not directly involved in binding to unmethylated H3K4. Some of the histone-binding domains within the RAI1 complex may play a role in the development of NDDs associated with RAI1. Missense mutations in individuals with SMS have been identified in both the ePHD and NBDs of RAI1, while nonsense or indel mutations associated with SMS lead to truncation of these domains.

Binding to unmethylated H3K4 acts as a searching mechanism to identify other unmethylated or recently demethylated H3K4 residues in the chromatin, specifically in gene promoters. Subsequently, the RAI1 complex recruits MLL1 to methylate H3K4, rendering the chromatin transcriptionally active. The action of the RAI1 complexes interacts antagonistically with LSD1–CoREST, establishing an opposing but balanced interaction that contributes to proper neuronal development. The LSD1–CoREST complex facilitates the demethylation of H3K4me1/2, repressing the expression of neuron-specific genes in neuroprogenitors. Through iBRAF, the RAI1 complex recruits MLL1 to methylate H3K4, thereby activating neuronal genes [7].

This intricate balance between the RAI1 and LSD1–CoREST complexes highlights RAI1′s crucial role in maintaining chromatin dynamics and regulating gene expression in the developing brain. The dysregulation of these mechanisms, particularly through mutations in RAI1, provides valuable insights into the molecular basis of neurodevelopmental disorders such as SMS. Understanding the precise mechanisms by which RAI1 modulates chromatin will be essential for developing targeted therapeutic strategies for these conditions.

#### 3.2.5. RAI1 Blocks Upscaling in Naive Networks and Promotes Inactivity-Induced Upscaling

The molecular function of RAI1 has been associated with its role as a transcription factor, playing a crucial and determinative role in the development and maintenance of the nervous system, particularly in synaptic plasticity. It has been shown that long-lasting forms of synaptic plasticity, such as synaptic scaling, critically depend on transcription [86] (Figure 5), requiring de novo synthesis of RNA and proteins that directly modulate synaptic efficacy. Notably, a recent study [4] describes the dual role of the RAI1 protein in regulating synaptic strength between neurons by blocking upscaling in naive networks and promoting inactivity-induced upscaling. RAI1 plays a crucial role in limiting synaptic upscaling in neural networks with baseline activity levels, preventing premature transcriptional responses that typically occur following changes in network activity. Several experimental findings support this model.

First, *RAI1* gene knockdown (RAI1 KD) was induced in mouse forebrain neuron cultures, showing a shift in gene expression towards transcriptional states typically associated with the presence of TTX, a toxin that blocks neuronal activity. This suggests that RAI1 is essential for maintaining a stable transcriptional state in neural networks with baseline activity. Furthermore, RAI1 KD leads to an increase in excitatory synaptic strength, as evidenced by the increased amplitudes of miniature excitatory postsynaptic currents and a rise in network firing rates. This phenomenon reflects a dysfunction in the balance of synaptic activity in the absence of RAI1 [7].

The second role of RAI1 is to promote synaptic upscaling when the network undergoes prolonged activity depression. Through its NBD, RAI1 binds to the nucleosome, acting as a transcriptional co-activator and potentially repressing the transcription of target genes. However, as of now, the genomic differences underlying these two functions remain unidentified. From previous studies, EHMT1/2 and TET3 are well-characterized chromatin regulators involved in synaptic scaling, and recently, RAI1 has also been shown to play a key role in homeostatic upscaling. It is possible that its dual functionality is mediated through interactions with these regulators within a complex [7].

Such an interaction becomes more plausible when we note that SMS and Kleefstra syndrome are often linked in differential diagnoses [87]. Additionally, Kleefstra syndrome and Beck–Fahrner syndrome are categorized under Disorders of the Epigenetic Machinery (MDEMs), reflecting the opposing yet complementary mechanisms of EHMT1 and TET3, respectively [88]. These clinical affinities could mirror a potential molecular interaction between RAI1, EHMT1/2, and TET3 in preserving neuronal network stability and regulating homeostatic plasticity, thereby reducing the risk of hyperexcitability or synaptic dysfunction, neurological features commonly observed in cognitive disorders associated with these conditions.

#### 3.2.6. Role of RAI1 in Modulating Age at Onset in SCA2 and Convergence with ATXN2-Associated Pathways

Some genetic neurodegenerative diseases are caused by the expansion of CAG trinucleotide repeats within the coding region of the target gene. These mutations result in the pathological elongation of a polyglutamine (polyQ) domain in the corresponding proteins, leading to their classification under the umbrella term polyQ diseases. A shared characteristic, although not universally observed, among all polyQ diseases is the inverse correlation between the length of the CAG repeat expansion and the age at clinical onset (age at onset, AAO). In other words, the mere presence of a pathogenic allele is not solely responsible for triggering clinical manifestations. Various factors can modulate this relationship, including cis- and trans-acting genetic modifiers, non-allelic variants, stochastic environmental influences, and potential sporadic events [89].

Spinocerebellar ataxia type 2 (SCA2) is one of the most prevalent autosomal dominant ataxias worldwide. It is caused by the expansion of a CAG repeat within the coding region of the ATXN2 gene, exceeding a pathological threshold of 31 repeats. However, approximately 50% of the variability in AAO among SCA2 patients remains unexplained [90].

Two main pathogenic mechanisms have been implicated in CAG repeat expansion disorders: toxic RNA-mediated gain-of-function effects, including RNA decay, and protein gain-of-function toxicity. Although the former is less extensively studied than protein toxicity, it nonetheless represents a significant pathogenic mechanism, primarily due to RNA-mediated toxicity. In such cases, the mutant RNA transcript forms abnormal secondary structures, such as hairpin loops or nuclear foci, that interfere with essential cellular processes, even in the absence of translation into a toxic protein. In contrast, protein toxicity is a well-established mechanism. Here, the expanded CAG repeat is translated into an abnormally long polyglutamine tract, resulting in structural and functional alterations of the protein. This promotes aberrant interactions with other cellular proteins. Notably, polyglutamine stretches are known to have a high propensity for self- and cross-interactions, which has led to the hypothesis that other proteins containing CAG repeats might interact with the expanded ATXN2 protein, affecting its aggregation kinetics and thereby modulating the age at disease onset [91].

One of these loci, *RAI1*, accounts for approximately 4.1% of the variance in the age of onset of SCA2, alongside other loci such as *ATXN3* and *CACNA1A*, in individuals with a confirmed pathogenic SCA2 allele [92]. Notably, although *RAI1* and *ATXN2* are involved in distinct genetic disorders, converging mechanisms related to lipid dysfunction have been observed. These include dysregulation of triglyceride levels and lipid droplet accumulation, alterations in sphingolipid and ganglioside metabolism with cerebellar and cerebral consequences, and the disruption of cholesterol synthesis, trafficking, or turnover, although these events occur through different mechanisms and in distinct cellular contexts [93]. RAI1 primarily acts as a transcriptional regulator in adipose tissue and the hypothalamus, whereas ATXN2 exerts its effects post-transcriptionally in cerebellar and cortical neurons [93,94].

Another functional axis shared by RAI1 and ATXN2 is the regulation of circadian rhythm. The RNA-binding protein ATXN2 binds and stabilizes numerous mRNA transcripts. Studies in Drosophila have demonstrated that ATXN2 modulates circadian rhythms through interactions with key clock proteins, such as PER and TIM, and promotes the post-transcriptional translation of specific mRNAs in pacemaker neurons. Furthermore, its role in cytoplasmic polyadenylation suggests a fine-tuned control over the timing of protein synthesis, a crucial aspect of circadian function.

Conversely, RAI1 acts at the transcriptional level, regulating core circadian genes such as *CLOCK* and *BMAL1*. This complementarity in regulatory levels suggests that both genes function as critical nodes within an integrated circadian network, whose disruption may compromise neuronal and metabolic homeostasis [95].

Although these functional connections remain to be fully elucidated, they may indicate converging roles in lipid homeostasis and circadian regulation, with potential relevance for therapeutic intervention.

## 4. Pathophysiological Mechanisms in RAI1 Haploinsufficiency

Understanding how cellular organization changes in the presence of RAI1 haploinsufficiency is crucial for delineating a detailed and comprehensive scenario of the pathophysiological mechanism (Figure 6). This knowledge is essential for identifying potential prognostic biomarkers that could be useful for assessing the effectiveness of targeted therapeutic treatments. Many studies have focused on fibroblasts rather than central nervous system (CNS) cells, where RAI1 is highly expressed and its haploinsufficiency has the most significant impact [10]. Williams et al., 2012 have tested the expression of genes involved in the CLOCK/BMAL1 pathway in SMS fibroblast cell lines and, concurrently, in the hypothalamus of RAI1+/− mice during both day and night phases. In both models, the same target genes were found to be downregulated [8]. This result is highly significant, as it demonstrates that SMS human fibroblast cell lines, despite not being the primary tissue target for assessing the pathogenic mechanisms of RAI1, represent a reliable cellular model for studying haploinsufficiency. This further underscores the importance of investigating the underlying molecular mechanisms to understand how the alterations observed in fibroblasts may reflect broader pathological processes and provide a foundation for future clinical applications.

### 4.1. Lipid Accumulation and Altered Lipid Metabolism

A previous study [9] used RAI1 haploinsufficient cell lines to identify genes with altered expression, employing approaches such as KEGG, GO, and RT-PCR. An increase in transcripts of genes associated with three main pathways, steroid biosynthesis (*SQLE*, *DHCR7*), lysosomal pathway (*ASAH1*, *HEXA*, *HEXB*, *LGMN*, *CTNS*, *IGF2R*, and *SLC17A5*), and protein trafficking (*SORT1*, *CLN3*, and *CTSH*) was observed. No significant variations were found among downregulated genes. These results suggest that RAI1 haploinsufficiency directly or indirectly affects the expression of key enzymes in these pathways.

The researchers also performed an untargeted lipidomic analysis to compare the lipid profiles of SMS cells with those of healthy control cells. They used LC–MS/MS to analyze lipids, collecting three biological replicates for each cell line. Principal component analysis revealed differences in lipid profiles. SMS cells showed significantly higher levels of triglycerides compared to controls, along with changes in ceramide and ganglioside levels. These lipid modifications were attributed to RAI1 haploinsufficiency, which alters triglycerides, sphingolipids, ceramides, and gangliosides, with a particular marked accumulation of triglycerides in SMS cells (Figure 6).

The same research group also analyzed lipid droplet (LD) accumulation in RAI1 haploinsufficient SMS cells and in healthy sibling cells. They used Oil Red O staining to quantify lipid incorporation. In SMS cells, the number of LDs increased 1.5- to 2.5-fold, and the size of the LDs was significantly larger compared to control cells. Furthermore, treatment with oleic acid (OA) stimulated lipid accumulation in control cells, but had no effect in SMS cells, suggesting that these cells are unable to produce additional LDs. These results demonstrate that cells with RAI1 haploinsufficiency accumulate LDs.

### 4.2. Autophagic Flux Disruption

After observing LD accumulation in SMS cells, the authors hypothesized that there could be a defect in autophagic flux, as LD accumulation is often associated with dysfunction in this process [96,97,98,99] (Figure 6). To test this hypothesis, they measured the levels of LC3-II and p62, two autophagy markers. Western blotting analysis showed increased levels of both LC3-II and p62 compared to control cells, suggesting an alteration in autophagy. The treatment of the cells with chloroquine, which blocks the fusion between autophagosomes and lysosomes, induced the accumulation of LC3-II and p62 in cells with normal autophagic flux. Additionally, LysoTracker staining showed a significantly stronger signal in SMS cells compared to control cells, indicating an accumulation of acidic organelles. These results suggest that RAI1 haploinsufficiency disrupts autophagic flux in SMS cells.

### 4.3. Mitochondrial Dysfunction and Oxidative Stress

In SMS cells with *RAI1* haploinsufficiency, ultrastructural analyses revealed a significantly higher number of cytosolic vacuoles with an empty appearance compared to control cells. Additionally, the mitochondria appeared empty, fragmented, and with abnormal cristae, and there was a significantly higher number of swollen mitochondria in SMS cells compared to control cells. These results suggest that the loss of RAI1 leads to the accumulation of abnormal vacuoles and alters mitochondrial morphology (Figure 6). Altered mitochondria are typically eliminated through a specialized form of autophagy known as mitophagy [99]. In line with the defects in autophagic flux, it was observed that the level of mitophagy was significantly lower in both SMS cell lines compared to control cells. These observations suggest that SMS cells are unable to eliminate depolarized mitochondria, likely due to a defect in autophagy.

The accumulation of LDs, defects in autophagy, and damaged mitochondria are often associated with the excessive production of ROS [100,101]. Using the fluorescent probe 2′,7′-dichlorodihydrofluorescein diacetate (DCFDA), ROS levels were significantly elevated compared to control cells. The accumulation of ROS can lead to cell death [102]. Through Trypan blue staining, it was observed that the percentage of dead cells increased and it was statistically significant, as confirmed by the TUNEL assay. These results suggest that the loss of RAI1 function leads to excessive ROS production, contributing to increased apoptosis and potential cell loss, which may represent a previously underappreciated aspect of SMS pathophysiology.

## 5. RAI1 and Smith–Magenis Syndrome Neurobiology

Brain development and function are exquisitely sensitive to *RAI1* copy number. *RAI1* is expressed across various brain cell types, with its expression initiating alongside the neuronal differentiation process. As reported in a previous work [103], *RAI1* is expressed in the branchial arch of the mouse embryo at embryonic day 9.5 (E9.5), which develops into craniofacial structures. In the E18.5 embryonic cortex, *RAI1* is found in postmitotic neurons but rarely in proliferating cells of the ventricular zone. This unique pattern is also observed in other developing brain areas, such as the dentate gyrus, cerebellum, and olfactory cortex. Additionally, *RAI1* was detected in a small fraction of S100B+ Bergmann glial cells in the cerebellum. The mRNA levels of *RAI1* increase during prenatal development, peak approximately one week after birth, and persist into adulthood in the murine model. RAI1 is widely expressed throughout the adult mouse brain and co-localizes with 78% of NeuN+ cortical neurons. RAI1 is expressed in both excitatory and inhibitory neurons in the thalamus and cortex. It is expressed in 75% of excitatory neurons that express Vglut1 (which encodes vesicular glutamate transporter 1) and in 57% of inhibitory neurons that express Gad1 and/or Gad2 (which encode glutamate decarboxylases) in the cortex [10]. RAI1 does not have a general housekeeping function required for every cell type; instead, it has more specific roles in certain cell types, which may imply that each SMS phenotype is caused by the loss of RAI1 in multiple non-overlapping cell lineages. Motor learning delay and obesity are key SMS-like phenotypes. Thanks to engineered murine models, Nestin^Cre^, RAI1^CKO^ was used to study the conditional expression of *RAI1*. The haploinsufficiency of RAI1 in Vglut2+ neurons is associated with motor delays, obesity, and, when RAI1 is absent, learning deficits (Figure 7A–C); meanwhile, in Gad2+ neurons, RAI1 loss leads to learning deficits (Figure 7B). RAI1 deficiency in Sim1+ cells, which include Vglut2+ PVH neurons, is a major contributor to overeating and obesity (Figure 7C) observed in Nestin^Cre^, RAI1^CKO^ mice. In contrast, RAI1 in SF1+ cells, including Vglut2+ VMH neurons, appears to play a less prominent role in regulating body weight [10].

### RAI1 and BDNF in SMS Cells: Disruption of Appetite Regulation

Among the hallmark clinical features of SMS is obesity, often linked to dysregulation in appetite control and energy balance. A valuable study shed light on the role of the *RAI1* gene in the PVH BDNF-expressing neurons and described the central role of the BDNF-TRKB signaling pathways in SMS pathogenesis [12] (Figure 8). BDNF is a neurotrophin that supports neuron survival, growth, and differentiation. In the hypothalamus, an area critical for regulating hunger and energy balance, BDNF plays an essential role in appetite control. RAI1 binds to an intronic region approximately 1 kilobase upstream of the activity-dependent BDNF promoter IV in HEK293 cells, suggesting a direct involvement of RAI1 in activity-dependent BDNF transcription [6,10]. *RAI1* depletion resulted in diminished BDNF expression in the mouse hypothalamus and in frog embryonic brains [44,104].

TRKB is the primary receptor through which BDNF exerts its effects on neurons [105]. Upon BDNF binding, TRKB activates several intracellular signaling pathways that modulate neuronal excitability and synaptic plasticity. The activation of the BDNF-TRKB pathway promotes satiety signaling and suppresses feeding behavior [106]; the proper functioning of this pathway is thus essential to maintain normal body weight by balancing food intake and energy expenditure. In SMS mice, this signaling pathway is disrupted. The study found that loss or reduction in RAI1 specifically in hypothalamic BDNF-producing neurons leads to decreased intrinsic excitability in these cells. This reduced activity may impair their ability to release sufficient BDNF or to respond effectively to BDNF via TRKB receptors. With impaired signaling at both the ligand and receptor levels, the regulatory mechanisms governing appetite become dysfunctional. As a result, affected individuals may experience hyperphagia, leading directly to obesity due to the failure to receive appropriate satiety signals or to match metabolic demand.

## 6. Gene Therapies and Alternative Treatment Approaches for SMS

Gene therapies and alternative treatment approaches for SMS offer new perspectives for addressing this complex and challenging condition. Early intervention is crucial, especially when the CNS is affected, as it is well established that neurogenic potential is closely linked to age [107]. Maximizing therapeutic outcomes through timely diagnosis and treatments is essential, particularly for disorders with neurodevelopmental and neurogenetic components, such as SMS. In this context, early diagnosis, potentially even prenatal, represents a key opportunity to implement effective therapeutic strategies. The challenges in treating neurodevelopmental disorders stem from their heterogeneous etiology and the difficulty of delivering therapeutic agents across the blood–brain barrier. Nevertheless, early treatment is critical not only to prevent irreversible changes but also to significantly improve therapeutic outcomes. Currently, the only medication administered to SMS patients is melatonin, used to manage sleep disorders and for its antioxidant properties [108]. To date, there is no cure for this syndrome, highlighting the urgent need to explore and develop innovative therapies, including gene-based and alternative treatments.

### 6.1. CRISPRa Enhancing Expression from the Endogenous RAI1 Functional Allele in a Tissue-Specific Manner

A valuable strategy to rescue haploinsufficiency phenotype is to increase endogenous gene expression using the CRISPRa system. In this approach, a catalytically inactive Cas9 (dCas9) is used to target a transcriptional activator to the gene’s regulatory element (such as the promoter or enhancer) [109,110]. A suitable single-guide RNA (sgRNA) is designed to guide dCas9 to the immediate vicinity of the gene’s regulatory element. The dCas9 enzyme is then fused to a transcriptional activation domain, such as VP64. When the dCas9–activator complex binds to the promoter, it stimulates the assembly of the transcriptional machinery and increases the production of messenger RNA from the target gene [111]. The CRISPRa system can overcome the problem of ectopic gene expression and may be particularly useful for genes that are not suitable for conventional gene therapy because their coding sequences exceed the packaging capacity of recombinant adeno-associated viruses (rAAVs). In the case of RAI1, this approach avoids off-target effects and is especially promising, as the size of the disease-causing gene exceeds the payload capacity of AAV vectors. Accordingly, a recent study explored the potential of CRISPRa-based therapy in a murine model, aiming to increase transcription from the remaining intact *RAI1* allele [10]. To achieve this, the researchers targeted a 500-nucleotide promoter region located immediately upstream of the RAI1 transcription start site. This region exhibits an open chromatin configuration in the mouse brain, making it accessible to CRISPRa-mediated activation. The authors tested the effectiveness of rAAV-CRISPRa in a cellular model using primary hippocampal neurons isolated from wild-type neonatal mice, obtaining encouraging and promising results: RAI1 mRNA levels increased approximately twofold; no alterations were observed in the expression of genes neighboring RAI1; RAI1 expression was significantly enhanced; several cell surface molecules known to be underexpressed in RAI1-deficient brains were upregulated; conversely, genes typically overexpressed in RAI1-deficient brains, such as *HSPG2*, *IGF2*, *NOS1*, *SLC16A1*, and *TTR*, were found to be downregulated. They then performed stereotaxic injection into the PVH of the hypothalamus in an SMS mouse model, as the neurobehavioral traits and obesity observed in SMS mice are caused by RAI1 deficiency in subcortical glutamatergic neurons, but not in cortical ones. This approach enabled targeted intervention while avoiding effects on the cortex. The authors found that endogenous RAI1 mRNA levels in the PVH of SMS mice were significantly elevated, BDNF mRNA levels increased and its expression in the PVH was partially restored, stereotypical repetitive behaviors were rescued, the treatment delayed the onset of obesity although it did not fully reverse it; the long-term expression of rAAV-CRISPRa did not trigger host immune responses nor elicit hepatic or local inflammation; and rAAV-CRISPRa did not significantly increase the expression of predicted off-target genes. Unfortunately, despite these positive outcomes, the rAAV-CRISPRa treatment did not improve social behavior deficits, a core feature of SMS. This finding strongly supports the notion that behavioral deficits in SMS mice involve multiple cell types across different brain regions. Therefore, it is plausible that such deficits can only be addressed by globally increasing RAI1 expression, not just in glutamatergic or GABAergic neurons. In this context, treatment with a new molecule such as SINEUP could represent a promising therapeutic avenue for SMS.

### 6.2. RNA Therapeutic Oligonucleotides: SINEUPs

In 2012, a group of researchers identified an innovative post-transcriptional regulatory mechanism called SINEUP, which has expanded the possibilities of RNA-based therapies. The term “SINEUP” combines SINE (Short Interspersed Nuclear Elements), repetitive sequences widely present in mammalian genomes, and UP, indicating mRNA translation activation. This discovery demonstrated how specific antisense lncRNAs containing SINEB2 repetitive sequences regulate the translation of protein-coding mRNA via a mechanism that does not rely on cap-dependent translation but instead occurs through direct interaction with ribosomes and translation complexes [112]. The researchers reached this discovery by studying lncRNAs that are typically located in the cell nucleus but move to the cytoplasm in response to specific signals, such as stress. Among these lncRNAs, the first and most extensively studied is antisense ubiquitin C-terminal Hydrolase L1 (*UCHL1*).

The activity of antisense *UCHL1* involves binding the sense mRNA of *UCHL1* in the cytoplasm and enhancing its translation, thereby promoting the synthesis of proteins crucial for cellular survival under stress conditions. *UCHL1*, in particular, is known for its role in neuroprotective mechanisms and cellular damage responses. What makes this discovery revolutionary is the ability of these RNAs to regulate translation without altering the amount of available mRNA, but rather by directly enhancing the translation process itself. This function has since been further validated and explored to better understand the underlying molecular mechanism. The inhibition of mammalian mTORC1, a major regulator of translation, has revealed how SINEUP can serve as an alternative strategy to maintain essential protein expression, such as *UCHL1*, under stress conditions when cap-dependent translation is compromised [113]. With this understanding, it is evident that SINEUP could represent a promising therapeutic approach for haploinsufficiency disorders, particularly SMS. This is due to their ability to increase RAI1 protein expression by approximately 1.5–3-fold, which is critical given the dosage sensitivity of the *RAI1* gene. Since their initial discovery in 2012, understanding of the SINEUP-mediated post-transcriptional regulatory mechanism has steadily progressed. While their activity was initially identified experimentally, significant advancements have been made in characterizing their structure. Today, the function of SINEUPS can be more precisely described by distinguishing two essential domains that regulate its activity: the binding domain (BD) and the effector domain (ED), which act at a post-transcriptional level. SINEUPs bind their target mRNA with high specificity via the BD, recruit them to heavy polysomes into the cytosol and stimulate translation via the ED, resulting in a ~1.5–3-fold increase in protein production. This mechanism may restore physiological protein levels in haploinsufficiency disorders while minimizing the risk of uncontrolled overexpression, ectopic expression, or genome instability. Furthermore, they are only active in tissues where the target mRNA is physiologically expressed. SINEUPs represent a compelling proof-of-concept strategy to reverse pathological phenotypes, addressing key limitations of traditional gene augmentation therapies and CRISPRa approaches.

Given these promising advantages in the current state of the art, three research projects are currently underway at different stages of development. In the first, SINEUP has been used to rescue frataxin levels in a cellular model of Friederich’s Ataxia [113]; in the second, involving in vivo model systems, SINEUP was shown to effectively rescue phenotypes associated with microphthalmia with linear skin defects in a medaka fish model of cox7b haploinsufficiency [114]. In the third, a SINEUP targeting *GDNF* mRNA was tested in a mouse model of Parkinson’s disease, leading to increased endogenous protein levels that persisted for at least six months [115]. A recently published study also demonstrated the ability of SINEUP RNA to rescue CHD8 haploinsufficiency, strongly associated with autism spectrum disorders, in multiple cellular and animal models. Specifically, the authors observed this rescue effect in patient-derived fibroblasts, human hiNPCs, mouse primary neuronal cells, mice, and zebrafish. These robust results provide a strong foundation for future clinical trial development [116].

### 6.3. Synthetic Drugs

#### 6.3.1. N-Acetylcysteine Modifies the SMS Cell Phenotype

Based on the important role played in several cellular mechanisms, previous studies have tested the efficacy of NAC, which has strong antioxidant capacity and regulates lipid metabolism [9], NAC reduced cell death and lipid accumulation. It is a thiol compound known for stimulating glutathione-S-transferase activity and acts as a free radical scavenger and antioxidant, by replenishing the intracellular pool of GSH. NAC has long been used to treat chronic obstructive pulmonary disease, but beyond its antioxidant and anti-inflammatory properties, there is increasing interest in its potential therapeutic benefits for metabolic disorders. Research has demonstrated that NAC supplementation can reduce hyperglycemia and hyperinsulinemia induced by high-fructose and high-sucrose diets, while also improving peripheral insulin sensitivity [117]. Given these results, NAC could represent a viable therapeutic option for SMS patients, particularly in managing the metabolic and cellular pathologies associated with the disorder. Future studies should focus on determining the optimal dosages, treatment durations, and potential combination therapies that could maximize the benefits of NAC. Additionally, the mechanisms by which NAC exerts these effects, particularly on lipid metabolism and cell survival, should be further explored to better understand its role in SMS and other related disorders. Furthermore, investigating NAC’s long-term safety and efficacy in preclinical models and clinical trials will be essential before considering its use in SMS treatment.

#### 6.3.2. Avenue Strategies Treating Obesity Targeting BDNF

The use of animal models of RAI1 deficiency has been essential for the identification and testing of a broad range of potential therapeutic targets, which include neurotransmitter systems, neuronal growth factors and related signaling pathways, and metabolic pathways. Although the outcome of any drug development program depends on multiple factors, such as study design and the quality of outcome measures, efforts in the field have led to the successful treatment of the SMS mouse model with LM22A-4. This small-molecule BDNF loop-domain mimetic acts as a selective modulator of the neurotrophin signaling pathways, particularly those involving BDNF and its receptor TRKB, which are known to be downregulated in the brains of SMS mice. Data show that LM22A-4 treatment significantly increased p-AKT levels in SMS mice, suggesting enhanced activation of the BDNF downstream signaling pathway. Furthermore, chronic treatment with LM22A-4 was shown to delay the onset of obesity in SMS mice (RAI1+/−). The compound also partially rescued disrupted lipid profiles and improved insulin tolerance, indicating beneficial effects on metabolic health. Additionally, it had positive effects on behavioral aspects related to stereotypical repetitive behaviors and led to a partial rescue of locomotor activity, which is also frequently altered in individuals with SMS. However, LM22A-4 did not significantly rescue the increased food intake, energy expenditure, or respiratory exchange rate in SMS mice.

For treating obesity, another recent work [118] found that PVH-specific BDNF overexpression during early adolescence fully reverses obesity in SMS mice, suggesting that postnatal manipulation of the BDNF signaling, either directly or indirectly, could be more effective than increasing RAI1. Undoubtedly, the encouraging results obtained with LM22A-4 treatment and increased BDNF expression in correcting obesity, as evaluated in SMS mouse models, when combined with future gene therapies and administered early, could lead to a significant reduction in the SMS phenotype and, at minimum, correct the main features of the disease.

## 7. Conclusions

Haploinsufficiency disorders represent a complex and multifaceted category of genetic diseases, characterized by the insufficient expression of a single functional allele, which can lead to a spectrum of developmental, metabolic, and neuropsychiatric abnormalities [17,119,120].

It is widely recognized that LCRs are embedded within pericentromeric and subtelomeric regions [121,122]. Their presence predisposes these regions to NAHR, resulting in genomic rearrangements. The 17p11.2 cytoband is flanked by seven LCRs, rendering it highly susceptible to meiotic rearrangements. Consequently, gametes may be formed with either a gain (17p11.2 duplication) or a loss (17p11.2 deletion) of genetic material, which are associated with PTLS and SMS, respectively. The 17p11.2 region harbors more than 80 genes, among which RAI1 has been identified as the major contributor to the phenotypes of these syndromes. *RAI1* is known to be dosage-sensitive and findings from both animal and cellular models underscore its role as a transcription factor, chromatin remodeler, and regulator of RNA quality control. These functions are essential for maintaining neuronal network integrity and ensuring proper gene expression during critical stages of brain development. Disruptions in RAI1 function, as seen in haploinsufficiency conditions, result in impaired synaptic scaling and widespread transcriptional dysregulation, which contribute to the neurobehavioral and metabolic abnormalities characteristic of SMS.

As highlighted in the literature, the current standard of care for SMS is primarily symptomatic, with interventions such as melatonin administration to address sleep disturbances; however, no definitive cure is available [123]. Here, we provided an overview of innovative strategies aimed at directly correcting RAI1 dysfunction or modulating its downstream pathways, including CRISPR-mediated gene activation. Small-molecule drugs, such as NAC and modulators like LM22A-4 have shown potential in restoring cellular morphology and metabolism. Additionally, RNA-based therapeutic oligonucleotides, such as SINEUP, represent a promising new frontier in the treatment of haploinsufficiency, as demonstrated by several research groups [15,113,115].

Therefore, a logical future direction would be to explore SINEUP as a therapeutic agent to enhance RAI1 protein levels. In conclusion, ongoing progress in the SMS and PTLS fields offers hope for addressing the challenges posed by these and other haploinsufficient disorders.

## Figures and Tables

**Figure 1 ijms-26-06667-f001:**
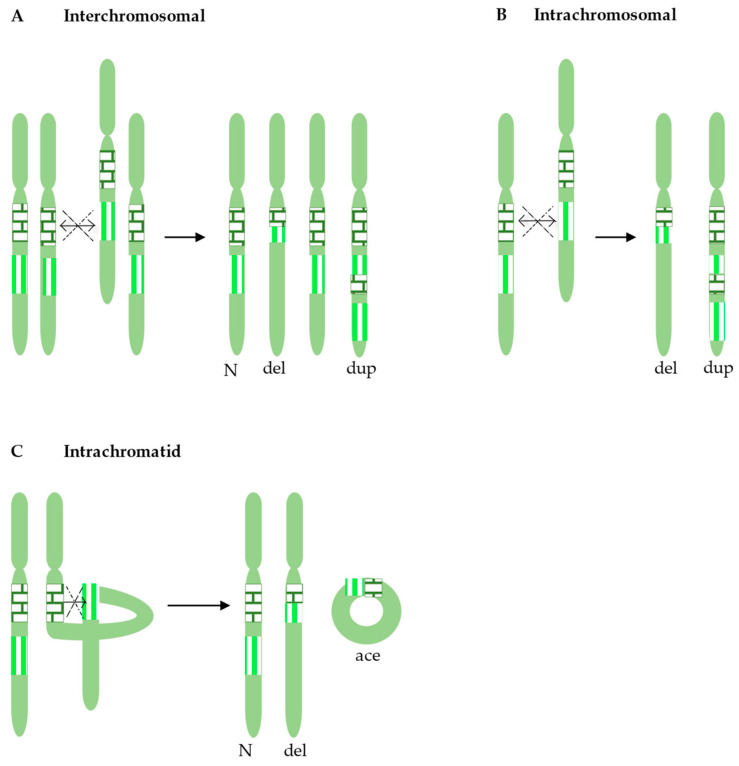
Genomic rearrangements by non-allelic homologous recombination (NAHR) between directly oriented LCRs. The figure illustrates the three different mechanisms by which structural genomic rearrangements can occur, leading to imbalances such as microdeletion, microduplication, and acentric chromosomes. (**A**) Interchromosomal NAHR. This schematic shows the potential NAHR between two homologous chromosome pairs. The presence of LCRs (striped and hatched) with high homology can lead to one allele slipping onto the other, resulting in a 1:1 unbalanced segregation of alleles. In this case, two chromosomes remain normal, while the other two comprise one that is microdeleted and the other microduplicated. (**B**) Intrachromosomal NAHR. The image shows recombination between two chromatids. For proper alignment of the LCR there is a possibility that the chromosome, instead of aligning correctly, may fold back on itself due to the high homology downstream of the LCR. This leads to the formation of two unbalanced chromosomes. (**C**) Intrachromatid NAHR. This image shows the final possibility, where NAHR occurs within the same chromatid. Due to the high homology of the LCR, the chromosome folds onto itself to facilitate the binding between the LCR located further downstream. This complex and rare structural rearrangement results in one normal chromosome, one carrying the microdeletion, and one acentric chromosome. N: normal; del: deletion; dup: duplication; ace: acentric.

**Figure 2 ijms-26-06667-f002:**
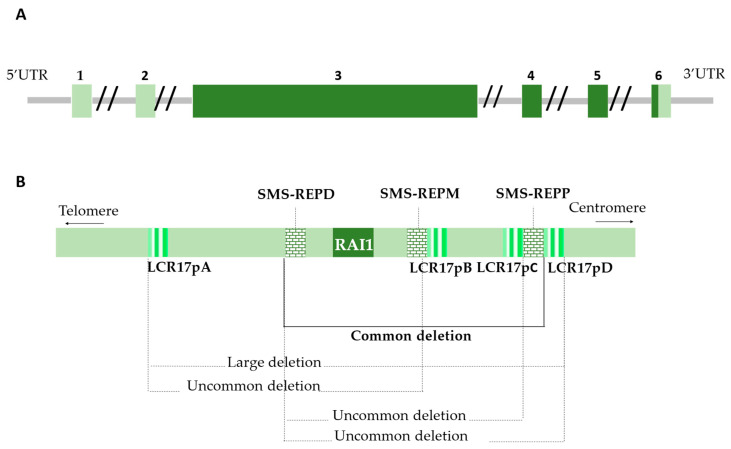
The genomic architecture of the *RAI1* gene (**A**) and chromosome region 17p11.2 (**B**) with the mechanisms underlying deletions leading to genomic imbalances. (**A**) A schematic representation of the human *RAI1* (OMIM #607642, NM_030665.4 GRCh38/hg38), which spans ~130 kb on chromosome 17, from positions 17,681,458 to 17,811,453 (UCSC Genome Browser). The coding exons (3, 4, 5, and part of 6) are shown in dark green, while the non-coding exons (1–2 and part of 6) are in light green. The 5′ and 3′ untranslated regions (UTRs) are represented by thick gray lines. Translation starts in exon 3, which encodes the majority (~97%) of the RAI1 protein. Exon 3 is also a mutational hotspot, often affected in RAI1-related disorders. (**B**) The genomic organization of the 17p11.2 region, illustrating the location of *RAI1* (green box) and the flanking low-copy repeats (LCRs; green and white stripes) and SMS–Repetitive Element (SMS-REPs: SMS-REPD, SMS-REPM, and SMS-REPP; shown as a brick-pattern motif; D = distal, M = middle, P = proximal). These highly homologous repeat elements mediate NAHR events, leading to recurrent or uncommon deletions. The most frequent rearrangement—the common deletion—occurs between SMS-REPD and SMS-REPP. Other deletion types (uncommon or large) involve recombination between different combinations of LCRs (LCR17pA–D). The orientation of the region is indicated from telomere to centromere.

**Figure 3 ijms-26-06667-f003:**
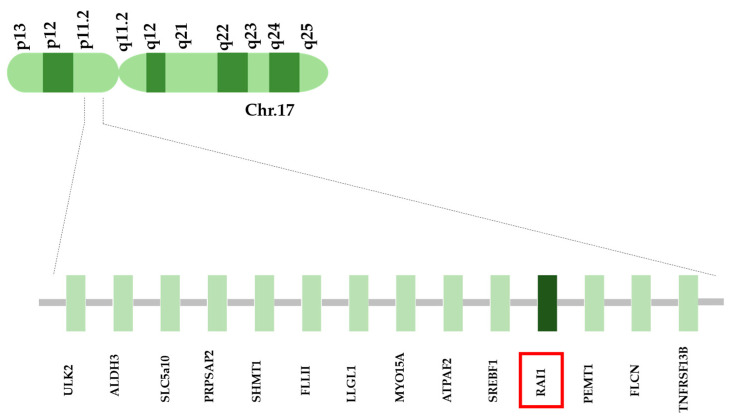
Diagrammatic representation of chromosome 17, highlighting the common deletion at the 17p11.2 locus. From left to right, the following elements are depicted: the G-band ideogram of human chromosome 17; and the schematic illustration of the Smith–Magenis syndrome region with OMIM related genes—emphasizing the nature of SMS as a contiguous gene syndrome and the contribution of multiple gene deletion to the complex and systemic phenotypic manifestations of the syndrome. The red box indicates the location of the RAI1 gene in relation to other genes within the 17p11.2 region.

**Figure 4 ijms-26-06667-f004:**
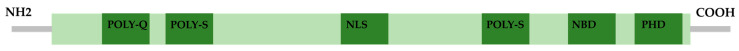
RAI1 Protein Structure. This figure illustrates the structural organization of the RAI1 protein, highlighting its several key functional domains: N-terminal polyglutamine-rich tract (Poly-Q), polyserine-rich domain (Poly-S), bipartite nuclear localization signal (NLS), sometimes considered as two distinct domains, second polyserine-rich tract (Poly-S), nucleosome-binding domain (NBD), and C-terminal plant homeodomain (PHD).

**Figure 5 ijms-26-06667-f005:**
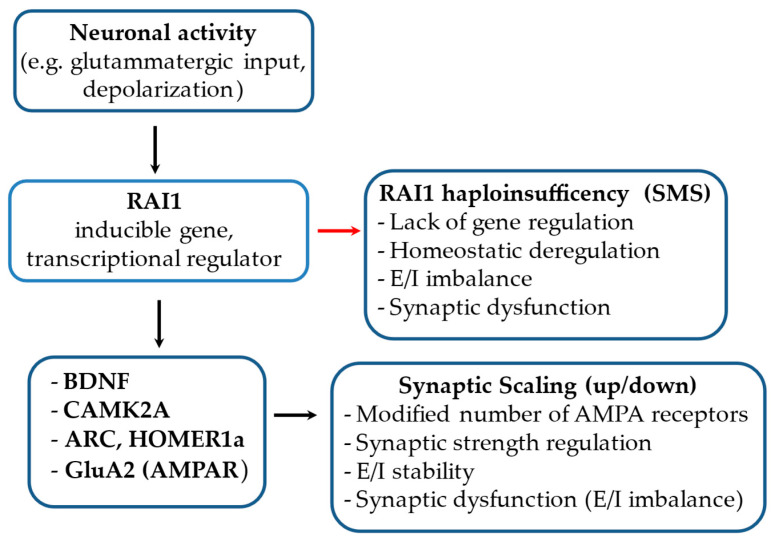
Diagram of the potential involvement of RAI1 in synaptic scaling. Neuronal activity induces the expression of RAI1, an activity-dependent transcriptional regulator that modulates genes involved in synaptic plasticity and the excitatory–inhibitory (E/I) balance, including BDNF, CAMK2A, ARC, HOMER1a, and GluA2. These targets contribute to synaptic scaling, a homeostatic mechanism that adjusts synaptic strength in response to prolonged changes in network activity. RAI1 haploinsufficiency may disrupt this regulatory pathway, leading to synaptic dysfunctions associated with neurodevelopmental disorders. The red arrow highlights the altered impact of RAI1 haploinsufficiency on synaptic scaling.

**Figure 6 ijms-26-06667-f006:**
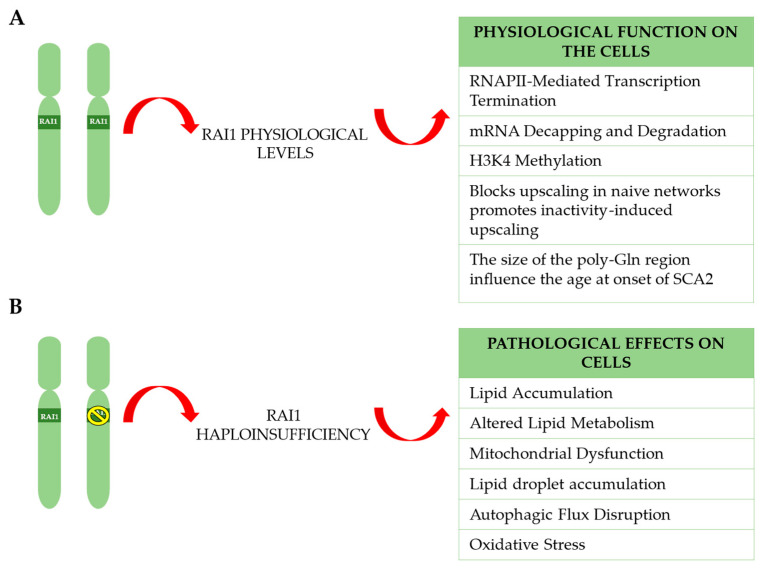
Schematic overview of RAI1 activity in physiological conditions and following haploinsufficiency. (**A**) Under physiological conditions, both alleles of the *RAI1* gene are expressed. RAI1 regulates multiple key cellular processes, including RNA polymerase II (RNAPII)-mediated transcription termination, mRNA decapping and degradation, and H3K4 methylation, all of which are essential for proper gene expression control. In neural networks, RAI1 blocks activity-independent upscaling and promotes inactivity-induced synaptic scaling. Additionally, a polymorphism in RAI1′s polyglutamine (poly-GLn) tract may modulate the age at onset in SCA2. (**B**) RAI1 haploinsufficiency disrupts these cellular functions, leading to a cascade of pathological cellular effects, particularly in neurons, including lipid accumulation, altered lipid metabolism, mitochondrial dysfunction, lipid droplet accumulation, autophagic flux disruption, and oxidative stress. The red arrows indicate the flow from genetic dosage to cellular outcomes in both physiological and pathological contexts.

**Figure 7 ijms-26-06667-f007:**
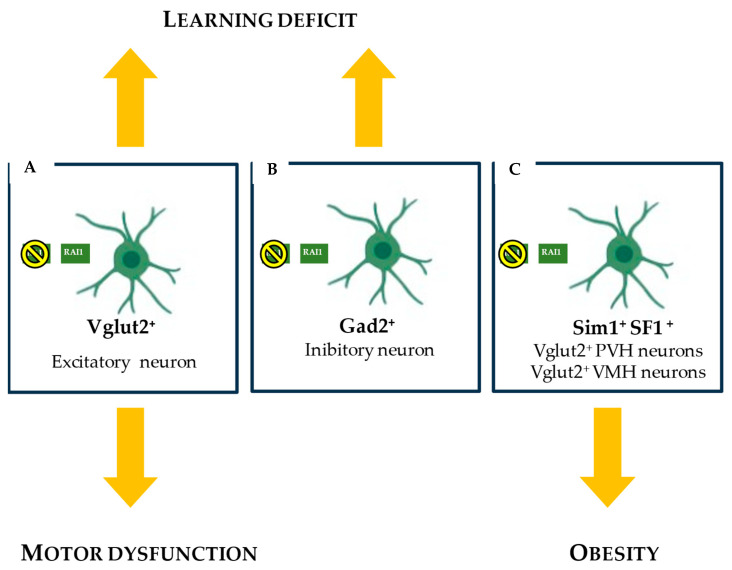
Neuronal roles of RAI1 in motor dysfunction and obesity in Smith–Magenis syndrome. This schematic illustrates RAI1 expression across key neuronal subtypes implicated in SMS phenotypes: excitatory Vglut2+ neurons, associated with motor delay, cognitive deficits, and obesity (**A**–**C**); inhibitory Gad2+ neurons (**B**), linked to cognitive deficits; and Sim1+ and SF1+ neurons located in the paraventricular (PVH) and ventromedial (VMH) hypothalamic nuclei, which are involved in appetite regulation (**C**).

**Figure 8 ijms-26-06667-f008:**
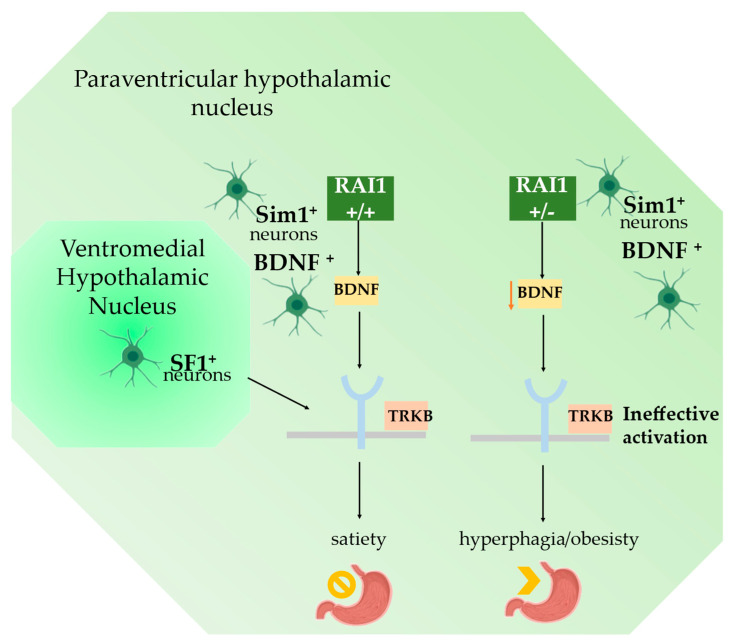
RAI1 deficiency impairs BDNF–TRKB signaling. Neuronal subtypes expressing or lacking RAI1 are shown. Under normal conditions, BDNF–TRKB signaling promotes satiety. In SMS, RAI1 deficiency leads to reduced BDNF levels, decreased TRKB activation, impaired satiety signaling, hyperphagia, and obesity. RAI1 binds to promoter IV of the BDNF gene.

## Data Availability

No new data were created or analyzed in this study. Data sharing is not applicable to this article.

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
