# Peer review of "Retinoic Acid Induced 1 and Smith–Magenis Syndrome: From Genetics to Biology and Possible Therapeutic Strategies"

_ijms, 2025, doi:10.3390/ijms26146667_

Round 1

Reviewer 1 Report

Comments and Suggestions for Authors

This manuscript provides a comprehensive and well-structured review of the role of the RAI1 gene in Smith-Magenis Syndrome (SMS) and Potocki-Lupski Syndrome (PTLS), covering genetic, molecular, clinical, and therapeutic aspects. The introduction effectively defines haploinsufficiency disorders and contextualizes RAI1 within this framework, although some redundancies between the abstract and introduction should be consolidated. The genomic architecture and NAHR mechanisms are clearly presented and well-supported. The section on RAI1's molecular functions—spanning transcription termination, RNA quality control, chromatin remodeling, and synaptic plasticity—is robust and up to date. The description of the gene's structure and functional domains is thorough, and the integration of findings from fibroblast and animal models is particularly strong, shedding light on disrupted metabolic pathways, autophagy, mitochondrial dysfunction, and oxidative stress. The neurobiological section effectively links RAI1 expression to brain circuitry, highlighting its cell-type-specific roles and behavioral outcomes in mouse models. The discussion of therapeutic strategies is a major strength, with detailed explanations of CRISPRa and SINEUP approaches. However, the pharmacological section, particularly on NAC and LM22A-4, would benefit from a more critical perspective. Figures, while informative, could be improved for clarity, especially Figures 2 and 5. Overall, the manuscript is scientifically rigorous and offers a valuable synthesis of RAI1-related mechanisms and emerging therapies, though minor revisions are needed to improve conciseness and critical depth.

Abstract: The sentence “This review highlights key molecular mechanisms of RAI1, elucidating its role in the interplay between genetics and phenotypic features and summarizes the innovative therapeutic approaches for SMS” is repeated and should be consolidated for clarity.

Additionally, the authors are encouraged to expand on the potential interactions between RAI1 and RNA-binding proteins such as ATXN2, a well-established modulator of neurodegenerative diseases. Notably, RAI1 has been implicated as a genetic modifier of age at onset in SCA2. Beyond this, there are functional overlaps between RAI1 and ataxin-2 in key regulatory pathways, including the CLOCK/BMAL1 circadian rhythm machinery and lipid metabolism, which could further enrich the translational relevance of the review.

Comments on the Quality of English Language

Please, authors may check thoroughly the English grammar of the manuscript. 

Reviewer 2 Report

Comments and Suggestions for Authors

This is a valuable review on the subject of Smith-Magenis and Potocki-Lupski syndromes  At present though there are many problems in grammar and explanations that make a significant proportion of the review difficult to follow.  Some of these are listed below but many grammatical mistakes exist in addition to those listed and the review will need careful revision with this in mind. Further a number of sections contain information that was already given in an earlier part of the review.  The review requires one person to go through and correct these consistently.  Some of these problems are listed below.

In the abstract the entire sentence starting on line 21 is repeated on line 23

 Lines 48-63 deal with the molecular events RAI1 regulates.  This list though finished on line 64 with a cellular event, synaptic plasticity.  This does not belong to this list and should be discussed separately (or the molecular steps regulating synaptic plasticity discussed)

Paragraph starting on line 77 is a confusing mish mash of genes regulated by RAI1 and molecular consequences.  Why are these put together in this paragraph? These are better discussed later with more explanation.

Line starting 117 “Moreover, considering… is very hard to follow and English needs to be improved.  Similarly line 125 starting “The origin of haploinsufficiency…” is difficult to understand the full meaning.   When saying “not mutually related one are…” I think this means a first reason, as the next sentence provides a second reason.  At present this is almost impossible to comprehend.

What is meant by “Microdeletion syndromes were first defined in 1998…” The genetics of Prader-Willi, Angelman syndromes and DiGeorge syndrome were determined in the 1980s.

Line 144 should be “Human genome sequencing…”

Figure 1 needs a key.  In the text it states the striped pattern is a LCR but does the different number of stripes in different examples mean something?  What does the hatched regions represent?

Line 234 “distinguishing and recognizing the specific clinical traits of that particular syndrome” I don’t think “recognising” is meant.

Line 235 Remove “m” from thatm regulate transcription,”

Starting line 270 “It is a dosage-sensitive gene implicated in several human disorders, including SMS and PTLS. These syndromes are caused by reciprocal microdeletions and microduplications of the same genomic region on chromosome 17p11.2.” introduces the subject as if the reader has not already read this.

Line 278 “Wide disparity, most SMS patients show hypercholesterolemia, elevated LDL levels, hyperphagia, and obesity by adolescence, emphasizing the crucial role of copy number variations in body weight regulation” reads as if some words are missing.

Line 281 states “Regarding the sleep cycle, the majority of SMS patients exhibit an inversion of melatonin cycle” , but nothing is actually said on how the sleep cycle changes.  This is repeated again on line 313 with the same problem! Overall there are multiple parts of the review repeating what was said earlier. 

Line 298 statement “until the genetic mutation is transmitted to the next generation, further highlighting the clinical subtleties of the syndrome.” needs to be explained.  Why might there be an effect in the next generation (and not earlier)?

Line 348 “RAI1 (initially named Gt1) was first discovered as a gene activated in the mouse embryonic tumor cell line P19 by retinoic acid, a treatment that induces the cells to develop into neuronal and glial cells.” What is the reference for this?

Line 364 In the sentences “These alternative transcripts might lead to different protein products with distinct regulatory roles, maintaining generally the same fundamental biological function. For example, some isoforms might have truncated versions of the RAI1 protein or lack specific domains present in the full-length form.”  are these facts or conjecture? If the latter, they are pointless as anything may be the case.

A figure on the biological functions of RAI1 would be useful to illustrate some of the steps of e.g. transcription termination and/or some of the proteins RAI1 interacts with in relation to its functions.  In addition, a figure of RAI1’s role in synaptic scaling would be useful – from the text its role in the feedback pathways necessary for this are not clear.

In figure 5 what do the curved arrows signify?

Section 4 finishes with the conclusion “These results suggest that the loss of RAI1 function leads to excessive ROS production and an increase in apoptosis.”  This implies that a significant part of the disorder consists of loss of cells.   This had not been made clear earlier in the review and, obviously, this must be an important aspect of this syndrome.

Why do the subtitles for sections 4 and 5 also include a reference to a figure (Fig 5)?  In addition Fig 5 does not provide much useful information for section 5.  Perhaps another figure can be designed for this.

Is the extensive section on SINEUP needed given the lack of studies on RAI1 using this?

Comments on the Quality of English Language

As above

Round 2

Reviewer 2 Report

Comments and Suggestions for Authors

My comments were nicely addressed and some relevant additional information added.